# The Role of the NRF2 Pathway in Maintaining and Improving Cognitive Function

**DOI:** 10.3390/biomedicines10082043

**Published:** 2022-08-21

**Authors:** Nora E. Gray, Marcelo Farina, Paolo Tucci, Luciano Saso

**Affiliations:** 1Department of Neurology, Oregon Health & Science University, Portland, OR 97239, USA; 2Department of Biochemistry, Federal University of Santa Catarina, Florianopolis 88040-900, SC, Brazil; 3Department of Clinical and Experimental Medicine, University of Foggia, 71122 Foggia, Italy; 4Department of Physiology and Pharmacology “Vittorio Erspamer”, Sapienza University of Rome, 00185 Rome, Italy

**Keywords:** NRF2 signaling pathway, cognitive decline, cognition improvement, neurodegenerative diseases

## Abstract

Nuclear factor (erythroid-derived 2)-like 2 (NRF2) is a redox-sensitive transcription factor that binds to the antioxidant response element consensus sequence, decreasing reactive oxygen species and regulating the transcription of a wide array of genes, including antioxidant and detoxifying enzymes, regulating genes involved in mitochondrial function and biogenesis. Moreover, NRF2 has been shown to directly regulate the expression of anti-inflammatory mediators reducing the expression of pro-inflammatory cytokines. In recent years, attention has turned to the role NRF2 plays in the brain in different diseases such Alzheimer’s disease, Parkinson’s disease, Huntington’s disease and others. This review focused on the evidence, derived in vitro, in vivo and from clinical trials, supporting a role for NRF2 activation in maintaining and improving cognitive function and how its activation can be used to elicit neuroprotection and lead to cognitive enhancement. The review also brings a critical discussion concerning the possible prophylactic and/or therapeutic use of NRF2 activators in treating cognitive impairment-related conditions.

## 1. Introduction

Nuclear factor (erythroid-derived 2)-like 2 (NRF2, also called NFE2L2) is a member of the cap‘n’collar subclass of the basic leucine zipper region containing the protein family. NRF2 is a redox-sensitive transcription factor that binds to the antioxidant response element (ARE) consensus sequence, regulating the transcription of a wide array of genes. The activity of NRF2 is tightly controlled by proteasomal degradation. Under normal conditions, NRF2 is sequestered in the cytosol bound to its cellular chaperone protein Kelch-like ECH-association protein 1 (KEAP1). When bound to KEAP1, NRF2 is targeted for degradation by the proteasome. However, in the presence of electrophiles or oxidative stress the nucleophilic cysteine sulfhydryl groups on KEAP1 are modified resulting in an allosteric conformational change that diminishes the KEAP-dependent degradation of NRF2 and allows the transcription factor to accumulate in the nucleus [1] (Figure 1). Nuclear translocation can also result from the phosphorylation of NRF2 at serine 40. Many kinases have been shown to phosphorylate this site on NRF2 in various tissue types including phosphoinositide-3 kinase)/protein kinase B (PI3K/AKT), mitogen-activated protein kinase (MAPK), extracellular signal-regulated kinase 1/2 (ERK1/2), glycogen synthase kinase 3 (GSK-3β) and protein kinase C (PKC) [2,3,4,5]. p62 and p21 can increase NRF2 transcriptional activity by decreasing the binding of NRF2 to KEAP1. p62 has been shown to physically block NRF2-Keap1 binding by itself binding to Keap1 in a location that overlaps the binding pocket for NRF2 [6]. In contrast, p21 directly interacts with the binding motifs of NRF2 and competitively inhibits KEAP1 binding [6].

The wide variety of ARE-containing genes include antioxidant and detoxifying enzymes such as gamma-glutamylcysteine synthetase, superoxide dismutase, catalase, glutathione reductase, thioredoxin reductase, peroxiredoxins, glutathione S-transferase and others [7]. NRF2 has also been implicated in regulating genes involved in both mitochondrial function and biogenesis [8,9]. NRF2 activation modulates the expression of ATP synthase subunit α and NDUFA4, two components of the electron transport chain (ETC) [10,11] as well as other enzymes important for proper bioenergetic function including malic enzyme 1, isocitrate dehydrogenase 1, glucose-6-phosphate dehydrogenase and 6-phosphogluconate-dehydrogenase [12,13]. The antioxidant effects of NRF2 also result in improved mitochondrial function, as decreasing the reactive oxygen species reduces oxidative damage to mitochondria. Additionally, NRF2 affects the expression of several regulators of mitochondrial biogenesis including sirtuin 1 (Sirt1), peroxisome proliferator-activated receptor γ (PPARγ) and PPARy co-activator 1 α (Pgc1α), considered to be the best regulator of biogenesis [14,15,16,17,18].

Significant cross talk also exists between antioxidant and anti-inflammatory pathways. For example, the classical pro-inflammatory transcription factor NF-κB is activated by oxidative stress which can be blocked by the NRF2-dependent induction of antioxidant target genes and thus the transcription of pro-inflammatory cytokines can be decreased [19,20,21]. However, NRF2 has been shown to directly regulate the expression of anti-inflammatory mediators such as IL17D, CD36, the macrophage receptor with collagenous structure (MARCO) and G-protein coupled receptor kinase (GRK) [22,23,24,25]. PPARγ is also known to have anti-inflammatory properties [26]. Moreover, NRF2 has been implicated in reducing the expression of the pro-inflammatory cytokines IL6 and IL1β through a binding site nearby the promoter that prevents the recruitment of RNA Polymerase II [27].

NRF2 is expressed in all tissue but in recent years attention has turned to its role in the brain and how its activation can be used to elicit neuroprotection and lead to cognitive enhancement [28,29]. Although experimental evidence points to an indisputable role of NRF2 activation in mitigating neurodegeneration and improving cognition, the clinical use of NRF2 activators for treating cognitive impairment related conditions represents an incipient and controversial research topic. The initial purpose of this review is to provide a comprehensive presentation and interpretation of the scientific literature supporting a role for NRF2 in maintaining and improving cognitive function. A second purpose of this review is to provide a critical analysis concerning the possible prophylactic and/or therapeutic use of NRF2 activators in treating cognitive impairment related conditions.

## 2. In Vitro Evidence

While it is not possible to test cognition per se in an in vitro system, cellular models can recapitulate the neuronal dysfunction as well as the synapse and dendrite loss that form the anatomic basis for cognitive decline [30] (Figure 2). There is a great deal of evidence from cellular models suggesting that NRF2 activation can ameliorate those endpoints.

Some models mimic the accumulation of pathogenetic amyloid beta (Aβ) observed in vivo that induces the accumulation of neurofibrillary tangles (NFTs), neuronal cell death and dementia [31]. 2,3′-dihydroxy-4′,6′-dimethoxychalcone (DCD), a member of the flavonoid chalcones group, as well as quercetin and rutin (present for example in citrus plants), syringin and eleutheroside B, have all been shown to be protective against Aβ-induced neuronal death through the activation of the NRF2/ARE pathway [32,33,34]. Similarly, tetra-butyl hydroquinone (tBHQ), a synthetic food preservative and NRF2 activator, and edaravone, a medication used to treat patients with amyotrophic lateral sclerosis, were likewise demonstrated to reduce oxidative stress, attenuate neuronal toxicity and reduce Aβ formation in an NT2N neuronal model of Alzheimer’s Disease [35] and SH-SY5Y neuroblastoma cells [36].

The water extract of *Centella asiatica* (CWE) has been reported to activate NRF2 in mouse primary neurons and protect against Aβ toxicity [37]. In neurons isolated from the Tg2576 mouse model of Aβ accumulation, CWE treatment was shown to reverse deficits in dendritic arborization and spine density [38]. A similar effect was observed in wild-type neurons and in MC65 neuroblastoma cells where CWE induced the expression of the antioxidant response gene NFE2L2 and its target genes to protect against Aβ-induced cytotoxicity [39].

CWE is phytochemically characterized by the presence of isoprenoids (sesquiterpenes, plant sterols, pentacyclic triterpenoids and saponins) and phenylpropanoid derivatives (eugenol derivatives, caffeoylquinic acids, and flavonoids) [40]. Several of these constituent compounds, including asiatic acid [41], madecassoside [42] and the 1,5-dicaffeoylquinic acid [43] have been shown to be neuroprotective and activate NRF2 in vitro as well.

Other phenolic compounds have also been reported to have beneficial effects in vitro that are linked to NRF2 activation. In PC12 cells, phenolic compounds induced neuronal differentiation and elicited neuroprotective effects through the binding of NRF2 [44].

The same effect was reported with curcumin, a natural polyphenol with multiple biological activities, including antioxidant and anti-inflammatory properties [45], and gypenoside XVII (a phytoestrogen isolated from *Gynostemma pentaphyllum,* belong to Cucurbitaceae family) [46,47]. To gain insights relevant for cognitive impairment and dementia, neurovascular models (composed of endothelial cells, myocytes, neurons, astrocytes, perivascular cells, microglia and oligodendroglia) have also been utilized [48]. The NRF2 activator sulforaphane, a metabolite of glucoraphanin (present in *Brassica oleracea*), reduced neuronal and endothelial death in primary brain endothelial cultures and maintained the integrity of the brain blood barrier (BBB) [49,50,51].

Another good correlate of cognitive function in vitro is synaptic density [52]. Different natural substances have been shown to induce overgrowth and neuroplasticity in vitro [53] and some of these are also known to activate the NRF2 pathway [54]. Carnosic acid, found in rosemary, which has been shown to have memory enhancing effects, has also been shown to induce neurite extension and neural differentiation through NRF2 activation in PC12h cells and the knockdown of NRF2 reduced this effect [55,56]. Similarly, in PC12 cells the flavonoid Luteolin stimulated neurite outgrowth (maximal neurite length and percentage of neurite bearing cells) in a dose-dependent manner [57].

## 3. In Vivo Evidence

There is ample evidence supporting the role of NRF2 in maintaining cognitive function in a variety of rodent model systems. This comes from the combination of studies demonstrating the deleterious consequences of loss of NRF2 and those detailing the cognitive enhancing effects of NRF2 activation (Figure 3).

### 3.1. Cognitive Impairing Effects of Loss of NRF2

Across a broad range of conditions, a reduction in NRF2 expression resulted in worsened cognitive outcomes. Mouse models of healthy aging have shown that NRF2 knockout (NRF2KO) mice experience accelerated cognitive decline between 6 and 18 months relative to wild-type (WT) and exacerbated cognitive impairments at older ages (17–24 months) [58,59,60].

Similarly, intensified cognitive impairment has been reported in mouse models of Alzheimer’s disease (AD) in which NFR2 has been knocked out. When NRF2 was deleted from the 5xFAD mouse model of Aβ accumulation, the expression of Aβ processing enzymes was enhanced leading to increased plaque pathology which was accompanied by worsened cognitive impairment as compared to 5xFAD animals that did express NRF2 [61]. This is consistent with earlier reports in mice that over express double mutations in amyloid precursor protein and presenilin 1 (APP/PS1 mice) in which NRF2 was ablated. In these animals, there was a significant exacerbation of deficits in spatial learning and memory along with working and associative memory relative to NRF2 expressing APP/PS1 mice [62,63]. Similarly, knocking out NRF2 in a mouse line modeling a combination of amyloidopathy and tauopathy together likewise worsened deficits in spatial learning and memory and also reduced long term potentiation in the perforant pathway [64].

The negative effects of the loss of NRF2 on cognitive function can also be seen in other neurodegenerative conditions. In a lipopolysaccharide (LPS) mouse model of multiple sclerosis, NRF2 deficiency resulted in more pronounced impairments in recognition memory [65] and for NRF2KO mice that were subjected to ischemic stroke showed an increased lesion volume and poorer neurocognitive performance [66].

### 3.2. Cognitive Enhancing Effects of NRF2 Activating Compounds

#### 3.2.1. Plant-Derived Compounds

A host of NRF2 activating compounds have demonstrated potent cognitive enhancing effects across a variety of models of cognitive impairment. The CWE has been shown to increase the expression of NRF2 and its regulated antioxidant target genes in vivo and to improve learning, memory and executive function in mouse models of both healthy aging as well as Aβ accumulation [67,68,69]. A number of other botanically derived compounds demonstrated similar NRF2-activating and cognitive enhancing effects. Pterostilbene (3′,5′-dimethoxy-4-stilbenol), found in berries and grapes, and curcumin have both been shown to activate NRF2 and improve cognitive function in mouse models of Aβ accumulation [70,71]. Sulforaphane has also been shown to reduce Aβ accumulation and ameliorate cognitive deficits in the 5xFAD mouse line as well as the 3xTgAD model of concurrent Aβ and tau accumulation [61]. It similarly improved cognitive impairment in mouse models of traumatic brain injury, diabetes and vascular cognitive impairment [51,72,73].

Incense, the resin of *Boswellia* genus plants, is used in religious ceremonies, as well as in medicine. The Boswellia extract contains triterpenoids that are pharmacologically active such as 3-O-acetyl-11-keto--boswellic acid (AKBA) and Boswellic acid (BA) that interact with NRF2 [74,75,76]. In APPswe/PS1dE9 mice, AKBA treatment improved performance in the Morris Water maze test of spatial and long-term memory, as well as in the passive avoidance task of fear-associated learning and memory in rodent models of central nervous system disorders [74,75,76]. Moreover, in APPswe/PS1dE9 mice treated with AKBA, the expression of nuclear NRF2 and total HO-1 were noticeably upregulated in the cerebral cortex and hippocampus while the NF-κB pathway was suppressed [76].

Medicinal mushrooms have also been shown to activate NRF2 and improve cognitive function. A recent study found that oral treatment with the medicinal mushrooms *Hericium erinaceus* and *Coriolus versicolor* reduced oxidative damage, neuroinflammation, and cognitive impairments in a mouse model of experimental traumatic brain injury. Male CD1 mice were subjected to a control cortical impact injury and then treated with 200 mg/kg of *Hericium erinaceus* and *Coriolus versicolor* for 30 days resulting in a potent induction of the expression of NRF2 and its antioxidant target genes in the brains of treated animals. This was accompanied by a reduction in NF-κB signaling as well as reduced markers of astrocytic, microglial activation and improved spatial learning and memory [77]. This study was in line with previous research which also showed that *Hericium erinaceus* and *Coriolus versicolor* activate NRF2, increase the expression of its downstream target genes [78,79] and improve cognitive function in mouse models of a high fat diet and Alzheimer’s disease [78,80].

Ginsenoside from ginseng (the root of plant *Panax ginseng*) and bilobalide and gingkolides from gingko (leaves of *Ginkgo biloba*) have likewise been reported to enhance cognitive performance in models of stress- and ischemia-induced cognitive impairment, respectively [81,82].

Lycopene, epigallocatechin gallate (EGCG) and resveratrol are also examples of the botanically derived NRF2 activating compound with reported cognitive enhancing effects in animal models of a wide range of conditions. Supplementation with lycopene (a carotenoid found in tomatoes, red fruits and vegetables) improved the cognitive function in rodent models of healthy aging and tauopathy as well as both high fat diet- and oxidative stress-induced cognitive impairment [83,84,85,86,87]. EGCG (a polyphenol highly abundant in green tea, with anti-oxidative and neuroprotective properties [88]) was similarly shown to improve high-fat diet induced cognitive deficits as well as those resulting from high-fructose consumption, Aβ overexpression and healthy aging [89,90,91,92]. Resveratrol is a plant-derived antioxidant polyphenolic compound that exhibits positive effects in animal models of neuropathological conditions [93,94]. Resveratrol attenuated cognitive impairment in a mouse model of traumatic brain injury and prevented type 2 diabetes-induced cognitive deficits [95,96].

#### 3.2.2. Synthetic NRF2 Activating Compounds

A number of synthetic compounds have also been identified as potent NRF2 activators with cognitive enhancing effects. The two synthetic compounds that have been most studied for their effects on cognitive performance are probably dimethyl fumarate (DMF) and tBHQ.

Fumaric acid esters, such as DMF and its primary metabolite monomethylfumarate (MMF), represent a class of molecules that have been shown to exhibit both antioxidant and anti-inflammatory properties [97]. Of note, there are studies showing the beneficial effects of DMF in counteracting oxidative stress and inflammation in neurodegenerative conditions through the activation of NRF2, although additional transcription factors (i.e., NF-κB) also modulated this compound [98]. DMF treatment was shown to improve spatial learning and memory in an experimental autoimmune encephalomyelitis mouse model of multiple sclerosis [99]. Spatial memory was likewise enhanced in aged rats treated with streptozotocin [100] and in a mouse model of subarachnoid hemorrhage [101]. Reference and working memory were also improved in those same mice [101]. DMF was reported to have similar cognitive enhancing properties in a model of sepsis [102], a double transgenic model of Aβ and tau accumulation [103] and a model of traumatic brain injury [104].

tBHQ is another small molecule that has shown promising cognitive enhancing effects. Rats treated with tBHQ showed reduced deficits in spatial learning and memory following subarachnoid hemorrhage [105]. tBHQ also improved spatial memory in rats, and spatial, associative and recognition memory in mice subjected to a mild traumatic brain injury [106,107].

### 3.3. Biochemical Pathways Associated with Cognitive Enhancement by NRF2 Activating Compounds

The interconnected and wide-ranging cellular consequences of NRF2 activation makes it difficult to identify a single mechanism of action underlying the cognitive enhancing effects of NRF2. In vivo effects on oxidative stress have frequently been reported alongside cognitive enhancement following treatment with NRF2-activating compounds as has improvement in mitochondrial function [37,59,63,69,70,76,85,86]. Activation of neuroprotective signaling pathways including ERK/CREB/BDNF, p38 MAPK/ERK and IRS/PI3K/AKT has also been implicated in mediating the cognitive effects of NRF2 activation [81,90,95].

Increased BBB permeability may also contribute to the beneficial effects of NRF2 activation on cognitive function although this has to our knowledge so far only been reported in the context of subarachnoid hemorrhage [92]. Another context-specific mechanism that may mediate additional cognitive enhancing effects could be alterations in beta and gamma secretase expression in Aβ overexpressing models [61,90].

The anti-inflammatory effects of NRF2 may also play an important role as neuroinflammation is a common feature of many conditions associated with cognitive impairment. Decreases in NF-κB signaling, reactive microgliosis, the overactivation of astrocytes and other markers of neuroinflammation have been reported where NRF2 activating compounds resulted in cognitive enhancement [63,65,76,83,86,87,90,103,104,107]. In Figure 4 a summary of the described biochemical pathways is provided.

## 4. Clinical Evidence for the Effects of NRF2 Activation on Cognitive Function

In line with the experimental studies supporting the role of NRF2 in maintaining cognitive function in animal models (discussed in the previous item), there is a growing body of evidence pointing to a potential relationship between NRF2 and cognition in humans. Particularly, there is mounting evidence indicating that NRF2 activators are able to improve (or at least prevent the decline of) cognition in humans. Nevertheless, although experimental studies with NRF2KO animals allow us to investigate the direct involvement of NRF2 in cognition, such a direct association is not as easy to determine in human studies. In the following sections, we provide an overview of the most frequently reported NRF2 activators with modulatory effects in cognition in humans.

### 4.1. Curcumin

Curcumin has been reported to display a favorable safety profile although recently a number of cases of acute non-infectious cholestatic hepatitis related to the consumption of curcumin dietary supplements were reported [108,109,110]. Curcumin is able to cross the BBB [111,112] without neurotoxicity, even at a high dose [113]. Curcumin activates NRF2 by alkylating a protein thiol on the Keap-1-NRF2 binding complex [114,115]. Concerning the effects of curcumin in human cognition, Ng et al. [116] observed that elderly subjects with occasional or frequent consumption of curry performed better than those reporting rare consumption.

Three years later, a randomized double-blind, placebo-controlled trial (ClinicalTrials.gov, accessed on 20 July 2022; Identifier NCT00164749) investigated the effect of curcumin in subjects affected by AD [117]. Thirty-four subjects were treated with different doses of oral curcumin (up to 4 g a day) or with placebo during 6 months. The authors found no significant effects of curcumin on cognition; however, they argued that the lack of cognitive decline in the placebo group may have precluded any ability to detect a relative protective effect of curcumin [117]. A second clinical trial (ClinicalTrials.gov, accessed on 20 July 2022; Identifier NCT00099710) evaluated the effects of Curcumin C3 Complex^®^ in 30 subjects with mild to moderate AD [118]. After 24-weeks of treatment, the authors were likewise unable to demonstrate clinical or biochemical evidence of the efficacy of the Curcumin C3 Complex(^®^).

Cox et al. [119] developed a study on the short-term effects of curcumin in a healthy, elderly population. After treatment with curcumin (solid lipid emulsion of curcumin named Longvida^®^ [120]), the authors found that both short term and chronic treatment with curcumin improved working memory and digit vigilance. In a similar study, Rainey-Smith et al. [121] performed a 12-month, randomized, placebo-controlled, double-blind study that investigated the ability of a curcumin formulation to prevent cognitive decline in a population of community-dwelling older adults. After six months of treatment, the placebo group manifested signs of cognitive decline that were not present in the group treated with curcumin.

More recently, Small et al. [122] studied the effect of curcumin (Theracurmin^®^ containing 90 mg of curcumin twice daily or placebo for 18 months) on memory in non-demented adults and explored its impact on brain amyloid and tau accumulation.

Cognitive skills (assessed by the Buschke Selective Reminding Test) were improved after six months and was maintained for the entire eighteen months of the trial. In a parallel group of subjects, curcumin induced a significant reduction in plaque and tangle deposition.

In 2020, Cox et al. [123] conducted a double-blind, placebo-controlled, parallel-groups trial in order to partially replicate their previous study [119]. Eighty aged participants (mean = 68.1 years) were randomized to receive Longvida© (400 mg daily containing 80 mg curcumin) or a matching placebo. The participants were assessed at baseline, and 4- and 12-weeks treatment. Compared with placebo, curcumin was associated with better working memory performance at 12-weeks, and lower fatigue scores at both 4- and 12-weeks; lower tension, anger, confusion and total mood disturbance were also observed in the curcumin group at 4-weeks only. The pattern of results is consistent with improvements in hippocampal function.

In summary, these clinical studies on the effects of curcumin in cognition/dementia do not provide a definitive response. As noted, there are studies reporting either beneficial or not significant results. It seems that the bioavailability of curcumin has a great influence in the achieved results. In addition, it is important to mention that the diverse protocols (measured cognitive skills) and characteristics of the enrolled subjects (especially age and health condition) in each of the aforementioned studies limits our ability to compare their results. Nevertheless, the reported beneficial effects of curcumin [122,123] indicate that this polyphenol represents a potential strategy to prevent the decline of cognition mainly in elderly individuals.

### 4.2. Centella asiatica

*Centella asiatica* has also been evaluated for its effects on human cognition. A double-blind trial developed more than four decades ago [124] showed significant increases in the general mental ability, attention and concentration of mentally retarded children treated with *Centella asiatica* (0.5 g/day during 6 months). Studies also reported improvements in cognitive function after *Centella asiatica* treatment in adult humans after stroke [125], as well as in healthy middle aged [126] and elderly individuals [127,128,129].

We found one clinical trial on the effects of *Centella asiatica* in humans with mild cognitive impairment (MCI) (ClinicalTrials.gov, accessed on 20 July 2022; Identifier NCT03937908). In summary, the study aimed to measure the oral bioavailability and pharmacokinetics of known bioactive compounds from a standardized *Centella asiatica* water extract product in mildly demented elders on cholinesterase inhibitor therapy. However, the trial was terminated before its conclusion.

### 4.3. Resveratrol

There are a significant number of clinical studies investigating the beneficial effects of resveratrol in human cognition. More than a decade ago, Kennedy et al. assessed the effects of oral resveratrol on cognitive performance and localized cerebral blood flow variables in healthy human adults [130]. In short, resveratrol administration resulted in dose-dependent increases in cerebral blood flow during task performance; however, cognitive function was not affected.

Evans et al. [131] tested whether chronic supplementation with resveratrol could improve cerebrovascular function, cognition and mood in post-menopausal women (aged 45–85 years).

Compared to placebo, resveratrol elicited improvements in the performance of cognitive tasks in the domain of verbal memory and in overall cognitive performance.

More recently, these same authors [132] developed a larger, longer term study to confirm the benefits of resveratrol in post-menopausal women. Compared to placebo, resveratrol improved overall cognitive performance and attenuated the decline in cerebrovascular responsiveness to cognitive stimuli.

Huhn et al. [133] performed a randomized controlled trial to determine the effects of resveratrol on memory performance and to identify potential related mechanisms using blood-based biomarkers, hippocampus connectivity and microstructure assessed with magnetic resonance imaging. Sixty elderly participants (60–79 years) were randomized to receive either resveratrol (200 mg/day) or placebo for 26 weeks (ClinicalTrials.gov, accessed on 20 July 2022: NCT02621554). This interventional study failed to show significant improvements in verbal memory after 6 months of resveratrol in healthy older adults. Similarly, in a study with sedentary and overweight older adults treated with placebo, 300 mg/day resveratrol, or 1000 mg/day resveratrol, Anton et al. [134] observed that 90 days of resveratrol supplementation at a high dose (1000/mg per day) selectively improved psychomotor speed but did not significantly affect other domains of cognitive function in older adults.

Despite the data indicating that resveratrol supplementation might improve select measures of cognitive performance [130,131,132], the current literature seems to be inconsistent and limited [135]. In a meta-analysis of 225 patients concerning the effect of resveratrol on cognitive and memory, Farzaei et al. [136] concluded that resveratrol has no significant impact on factors related to memory and cognitive performance. More randomized controlled trials are needed to achieve more conclusive results.

### 4.4. Sulforaphane

We identified two recent studies concerning the effects of sulforaphane in human cognition. Liu et al. [137] are developing a double-blind randomized controlled clinical trial to assess the efficacy of sulforaphane for improving cognitive function in patients with frontal brain damage. In this trial (ClinicalTrials.gov, accessed on 20 July 2022: NCT04252261), ninety eligible patients (having cognitive deficits after frontal brain damage) will be randomly allocated to sulforaphane treatment or placebo; they will undergo a series of cognitive and neuropsychiatric tests at baseline at different time points to determine the effect of sulforaphane on cognition. This study will also evaluate brain metabolites markers (including N-acetyl aspartate, glutamate, glutathione and γ-aminobutyric acid) and long-term outcomes of brain trauma, brain tumors and cerebrovascular disease via exploratory analyses. In July 2022, the recruitment status was “Not yet recruiting”.

A very recent clinical trial by Nouchi et al. [138] examined whether combined brain-training (BT) and sulforaphane intake intervention has beneficial effects on cognitive function in older adults.

The authors observed that the BT and sulforaphane treatments separately led to improvements in cognitive functions; the BT group showed a significant improvement in processing speed compared to the active intervention group and the sulforaphane intake groups revealed significant improvements in processing and working memory performance compared to the placebo supplement intake group.

Moreover, sulforaphane administered to 10 patients with schizophrenia induced an improvement in cognitive function, which was evaluated using the Japanese version of CogState battery at the beginning of the study and after 8-weeks of treatment [139]. We also found one ongoing clinical trial on the effects of sulforaphane on cognition in prodromal to mild AD (ClinicalTrials.gov, accessed on 20 July 2022; Identifier NCT04213391). The investigators will evaluate the efficacy, safety and related mechanism of sulforaphane in the treatment of AD patients. The study will recruit 160 AD patients, and then these patients will be randomized to a sulforaphane group or placebo group (80 patients per arm) for a 24-weeks clinic trial. The Alzheimer’s Disease Assessment Scale (ADAS-cog) will be performed at screen/baseline, 4-weeks, 12-weeks and 24-weeks to test the cognition of patients. As of July 2022, the recruitment status was “Recruiting”.

### 4.5. Epigallocatechin Gallate

We found three clinical studies on the effects of EGCG on human cognition. In 2012, Scholey et al. [140] investigated whether EGCG modulates brain activity and self-reported mood in a double-blind, placebo controlled crossover study. In short, at baseline assessments or at 120 min following the administration of 300 mg EGCG or matched placebo, cognitive and cardiovascular functioning, mood and a resting state electroencephalogram (EEG) were evaluated. The authors observed that EGCG administration significantly increased alpha, beta and theta activity, which was also reflected in overall EEG activity. In comparison to the placebo, the EGCG treatment also increased self-rated calmness and reduced self-rated stress, suggesting that participants receiving EGCG may have been in a more relaxed and attentive state after consuming the compound. It is important to stress that this attentive state was detected after acute EGCG treatment, suggesting that EGCG may act as a potential cognitive enhancer.

Approximately 5 years ago, de la Torre et al. [141] developed a double-blind, randomized, placebo-controlled, phase 2 trial to evaluate whether the administration of a green tea extract containing EGCG would improve the effects of non-pharmacological cognitive rehabilitation in young adults with Down’s syndrome (ClinicalTrials.gov, accessed on 20 July 2022; NCT01699711). After 12 months, EGCG and cognitive training was significantly more effective than placebo and cognitive training at improving visual recognition memory, inhibitory control, and adaptive behavior. No differences were noted in adverse effects between the two treatment groups. The authors state that Phase 3 trials with a larger population of individuals with Down’s syndrome will be needed to assess and confirm the long-term efficacy of EGCG and cognitive training.

Liu et al. [142] performed a single-blind, placebo-controlled, crossover study to examine the effects of green tea extract on working memory in healthy younger (21–29 years) and older (50–63 years) women. The study was characterized by a small number of subjects, whereby twenty non-smoking Caucasian women were recruited in the younger (10) and older (10) age group. Subjects received 5.4 g of green tea extract (at least 45% EGCG) or placebo within a 24-h period. Green tea extract significantly improved reading span performance in older women (higher absolute and partial scores of reading span), although no significant changes were observed in the younger group.

Among the three aforementioned studies concerning the effects of EGCG in human cognition, two of them suggested that short-term EGCG treatment can display significant effects in cognition, leading to an attentive state [140] and improving reading span performance [142]. Concerning the study by de la Torre et al. [141], the assumed beneficial effects of EGCG (combined with cognitive training) in visual recognition memory, inhibitory control, and adaptive behavior of young adults with Down’s syndrome seemed to open a new therapeutic possibility, although additional trials with a larger population of individuals with Down’s syndrome will be needed to better assess this issue.

### 4.6. Dimethylfumarate

In 2013, DMF was approved by FDA to treat patients with relapsing forms of multiple sclerosis (MS) [143]. Despite the significant amount of data showing that DMF is able to improve (or prevent the decline of) cognition in experimental animal models, we found only one clinical study concerning DMF and cognitive function in humans.

Amato et al. [144] performed a clinical study to evaluate the effect of 2-year treatment with oral DMF on cognition in relapsing remitting MS (RRMS).

In short, the authors concluded that the 2-years treatment with DMF was associated with the slowing of cognitive impairment and with significant improvements in quality of life and psychosocial function. Based on the mechanism of action of DMF, as well as on the fact that early and progressive cognitive decline in patients with MS represents a consequence of an immune-mediated inflammatory condition leading to demyelination and synaptic loss [145], the results of Amato et al. [144] strongly suggest that DMF was able to mitigate the predicted cognitive decline resulting from neuroinflammatory and neurodegenerative events that occur in MS, rather than acting as a cognitive enhancer.

### 4.7. Extract of Boswellia Species

MS patients were recruited in a randomized, double-blinded, and placebo-controlled study (IRCT2013070813911N1), to evaluate the effects of *Boswellia papyrifera* administration (300 mg, twice daily per os) on visuospatial and verbal memory and also information processing speed. After 2 months, only visuospatial memory was improved in patients treated in comparison to placebo [146]. However, the trial suffered from several weaknesses including that it involved only 38 patients and neither the chemical composition of *Boswellia papyrifera* with information on the percentage of the AKBA and BA nor a description of capsule production were reported.

## 5. Potential Use of NRF2 Activators to Treat Cognitive Impairment Related Conditions: Drawbacks and Perspectives

In this review, we summarized the substances used in attempts to stimulate NRF2 in different in vitro and in vivo models. Although promising, only some of the beneficial effects of these agents were detected in clinical testing (Table 1).

Cognitive deficits resulting from neurodegenerative conditions (commonly characterized by atrophy and/or loss of neuronal cell function) are unlikely to recover unless significant neuroplasticity or neurogenesis occurs. Considering that NRF2 is a transcription factor that orchestrates the expression of cytoprotective genes [147] and that mRNA and protein synthesis represent events dependent on adequate cellular metabolic homeostasis, which is compromised in neurodegenerating neurons, the potential therapeutic use of NRF2 activators to treat cognitive impairment resulting from neurodegenerative conditions appears to be unlikely. In this regard, although there is a growing body of experimental evidence pointing to NRF2 activation as a strategy to combat neurodegeneration and improve cognition (see Section 3), it appears that the reported beneficial effects of NRF2 activators in these studies primarily result from the prevention of neurotoxicity, neuroinflammation and/or neurodegeneration. According to this perspective, to the best of our knowledge, there is no scientific evidence pointing to the beneficial clinical effects of NRF2 activators in improving cognitive deficits in patients with neurodegenerative diseases.

When reviewing some reflections related to the available clinical literature on the effects of NRF2 activators in human cognition, it is important to mention that cognitive decline, which represents an expected consequence even among relatively healthy “successful agers”, is associated with structural, functional, and metabolic brain changes whose mechanisms involve oxidative stress and neuroinflammation [148]. Moreover, cognitive decline is commonly more intense in dementia frontotemporal and with Lewy bodies but also in individuals with specific diseases, such Alzheimer’s disease [149], Parkinson’s disease [150,151], type 2 diabetes mellitus [152], and Huntington’s disease [153], among others. Based on the well-known antioxidant and anti-inflammatory events resulting from NRF2 activation [154], some of the aforementioned clinical evidence on the beneficial effects of NRF2 in human cognition strongly suggest that the described NRF2-based interventions were able to prevent cognitive decline rather than directly improve cognition.

In this context, it is important to note the clear distinction between cognitive enhancers and disease modifying strategies [155]. Particularly in the clinical studies involving patients (MS, mental retardation, Down’s syndrome) [124,141,144] or elderly subjects [116,119,122,126,127,128,129,131,132,138], it is likely that, due to the antioxidant and anti-inflammatory effects resulting from NRF2-related downstream proteins, the upregulation of this pathway prevented cognitive decline directly by preventing neurodegeneration, thus displaying an indirect effect in promoting cognition.

On the other hand, some clinical studies observed improvements in cognition even after acute treatments with NRF2 activators (particularly, EGCG) [140,142], pointing to the direct effects of this compound in human cognition (not necessarily linked to the prevention of neurodegenerative events). Considering the acute effect of EGCG on cognitive ability, one may posit that this compound directly enhances cognition in humans. Even though this hypothesis seems reasonable, the available studies [140,142] do not allow for the conclusion that such improvements in cognition were specifically the result of NRF2-mediating events. In fact, the direct acute effect of EGCG in modulating the neurotransmitter/synaptic homeostasis on cognition via NRF2-independent mechanisms cannot be ruled out [156]. In conclusion, the summarized in vivo and clinical studies suggest that some NRF2 activators (described in Section 3 and Section 4) can improve cognition, especially under specific neuropathological conditions, due to their ability to prevent neurodegeneration. However, the potential clinical use of NRF2 activators to treat cognitive impairment after neurodegeneration has occurred appears to be unlikely.

## Figures and Tables

**Figure 1 biomedicines-10-02043-f001:**
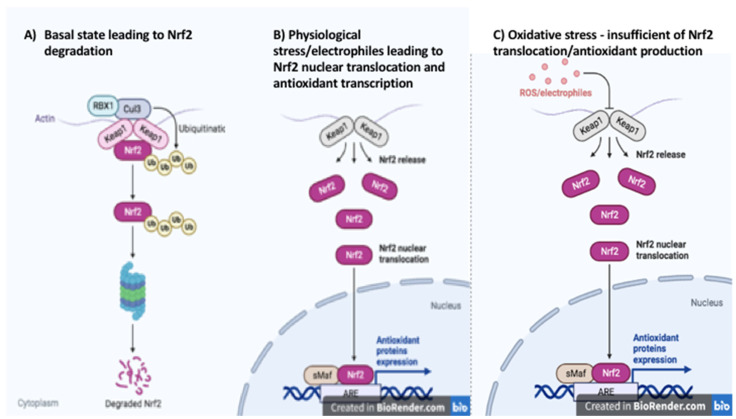
Keap1–Nrf2 Pathway. (**A**) basal state leading to Nrf2 degradation, (**B**) physiological stress/electrophiles leading to Nrf2 nuclear translocation and antioxidant transcription, (**C**) oxidative stress—insufficient of Nrf2 translocation/antioxidant production (created in BioRender.com, accessed on 20 July 2022).

**Figure 2 biomedicines-10-02043-f002:**
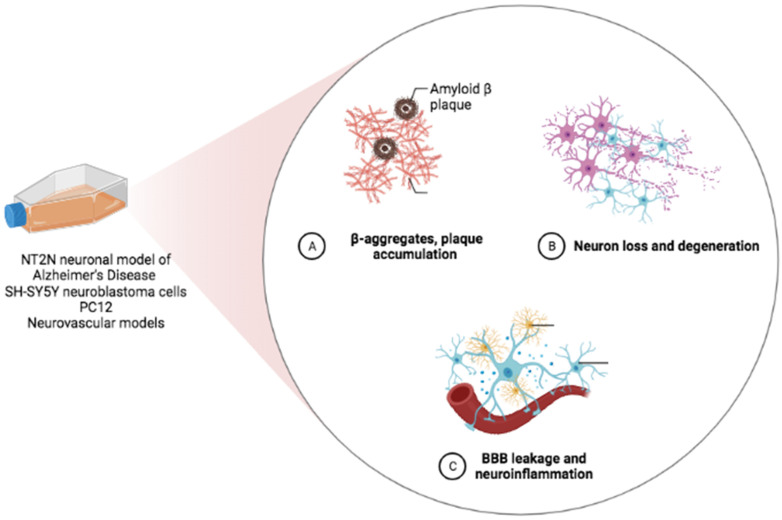
In vitro models recapitulating the neuronal dysfunction, synapse and dendrite loss that form the anatomic basis for cognitive decline (created in BioRender.com, accessed on 20 July 2022).

**Figure 3 biomedicines-10-02043-f003:**
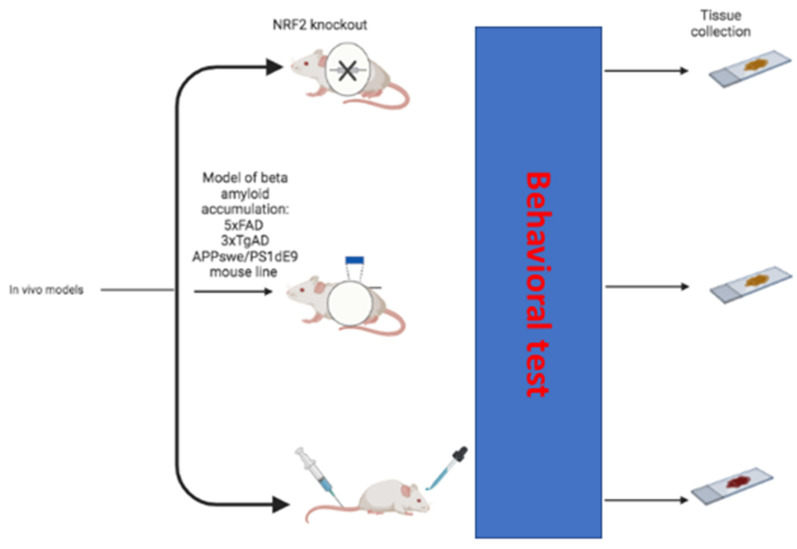
In vivo models used to demonstrate the deleterious consequences of loss of NRF2 and the cognitive enhancing effects of NRF2 activation (created in BioRender.com, accessed on 20 July 2022).

**Figure 4 biomedicines-10-02043-f004:**
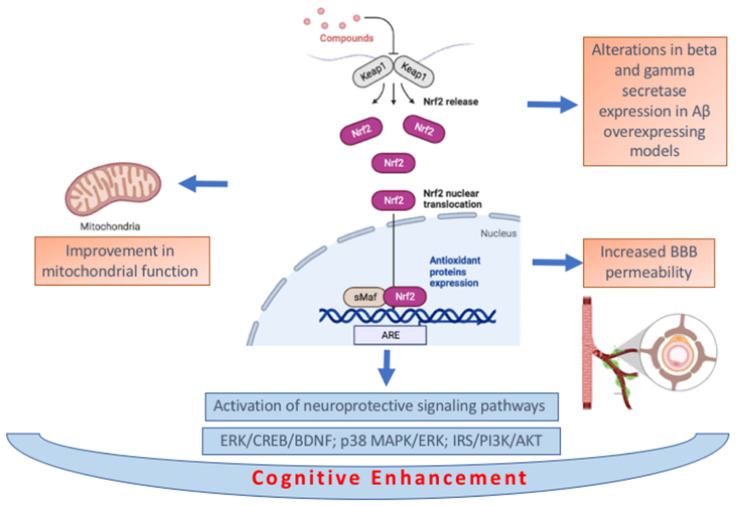
Summary of biochemical pathways associated with cognitive. enhancement by NRF2 activating compounds (created in BioRender.com, accessed on 20 July 2022).

**Table 1 biomedicines-10-02043-t001:** Evidence from substances treatment supporting a role for NRF2 in maintaining and improving cognitive function. In this table are reported a summary of results from substance tested in vitro and/or in vivo and in clinical trials.

Compound	In Vitro	In Vivo	Clinical Trial
Curcumin	Induction of neuronal differentiation and neuroprotective effects in PC12 cells [45].	Improvement of cognitive function in mouse models of Aβ accumulation [70,71].	Conflicting results [116,117,118,119,120,121,122,123].
*Centella asiatica*	Reverse deficits in dendritic arborization and spine density in neurons isolated from the Tg2576 mouse model of Aβ accumulation, in wild-type neurons and in MC65 neuroblastoma cells [38,39].	Improvement of learning, memory and executive function in mouse models of both healthy aging and Aβ accumulation [67,68,69].	Increase in the general mental ability, attention and concentration of mentally retarded children. Improvement in adult humans after stroke, in healthy middle age and elderly individuals [124,125,126,127,128,129]. ClinicalTrials.gov, accessed on 20 July 2022: NCT03937908.
Resveratrol	None	Attenuation of cognitive impairment in a mouse models of traumatic brain injury [95,96].	Data are inconsistent and limited [130,131,132,133,134,135,136].
Sulforaphane	Neuronal and endothelial death reduction in primary brain endothelial cultures [49,50,51].	Reduction of Aβ accumulation and amelioration of cognitive deficits in the 5xFAD and the 3xTgAD mouse line [51,61,72,73].	Data are inconsistent and limited [137,138,139].
Epigallocatechin gallate	None	Improvement of cognitive deficits induced from Aβ overexpression mouse model, high-fat diet or high-fructose consumption. Improvement of cognitive deficits also in healthy aging [89,90,91,92].	Three clinical studies. Short-term treatment improved cognition. Beneficial effects in visual recognition memory, inhibitory control, and adaptive behavior of young adults with Down’s syndrome [140,141,142].
Dimethylfumarate	None	Improvement of cognitive performances in autoimmune encephalomyelitis mouse model of multiple sclerosis, in aged rats treated with streptozotocin, in a mouse model of subarachnoid hemorrhage, in a model of sepsis, in a double transgenic model of Aβ and tau accumulation and in a model of traumatic brain injury [99,100,101,102,103,104].	Only one clinical study. The 2-year treatment was associated with slowing of cognitive impairment [144].
Extract Boswellia	None	Alleviation of cognitive deficiencies in APPswe/PS1dE9 mice (AKBA treatment) [76].	One clinical study [146]. Data are inconsistent and limited.

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
