# Peer review of "The Role of the NRF2 Pathway in Maintaining and Improving Cognitive Function"

_biomedicines, 2022, doi:10.3390/biomedicines10082043_

Round 1

Reviewer 1 Report

The submission from Nora Gray et al. reports the role of NRF2 activation in maintaining and improving cognitive function and how this pathway can be used to elicit neuroprotection and lead to cognitive enhancement. The review is interesting and well structured. However, there are some corrections.

Minor comments:

1.     The authors should better highlight the purpose of the review and the novelty in the introduction section

2. Authors should write the entire name first and then use the acronyms indicated in parentheses.

Example: lines 30-31 correct KEAP1 (Kelch-like ECH-association protein 1) with Kelch-like ECH-association protein 1 (KEAP1).

3. Aurtors should include among the preclinical studies those in which the natural compounds Hidrox, Hericium herinaceus and Coriolus versicolor were used which have been shown to activate or modulate the NRF2 pathway (10.3390/antiox9090824; 10.3390/ijms21113893; 10.1155/2018/5802634; 10.3390/antiox10060898)

4.     The authors should better check the manuscript for any typographical errors.

Reviewer 2 Report

 1.This review needs revision. While the review discusses the role of NRF2 pathway in cognitive function, it is not innovative enough and quite descriptive. There are similar reviews. The name is called “Targeting Transcription Factor Nrf2 (Nuclear Factor Erythroid 2-Related Factor 2) for the Intervention of Vascular Cognitive Impairment and Dementia”. The whole review is narrating and does not refine a new point of view.

2.     This review only one table and one figure, please increase the number of charts.

3.     Line 494,cognitive impairment is more common in dementia with Lewy bodies, frontotemporal dementia than in type 2 diabetes?

4.    Line125,what cell is PC12h? Does it have an extra h? Is Typos?

5.     It is suggested to write references in the Table1.

6.     Line177-180,Line344-349 What references were cited. In addition, the content is lengthy and needs to be concise.

7.     Line160å’Œ201 It is suggested to add a numeric serial number before the subtitle, which is more appropriate. 

8.   The author did not examine the review carefully and made a low-level error. Line 278 (Baum 2008) and Line 332 (Farhana 2016) is superfluous.

Round 2

Reviewer 2 Report

recommended accept.